# Justice System Contact and Health: Do Immigrants Fair Better or Worse than the Native-Born after Arrest, Probation, or Incarceration?

Casey T. Harris *, Michael Nino, Zhe (Meredith) Zhang and Mia Robert

Department of Sociology and Criminology, University of Arkansas, Fayetteville, AR 72701, USA
* Correspondence: caseyh@uark.edu; Tel.: +1-479-575-3205

**Abstract:** Despite decades of both macro- and micro-level studies showing immigration to be unassociated or negatively linked to crime, research examining the consequences of justice system contact among immigrants has been comparatively underdeveloped. The current study examines whether justice system contact (arrest, probation, and incarceration) is linked to poorer health and, in turn, whether there were differences in how justice system contact is related to immigrant versus native-born health. Using data from multiple waves of the National Longitudinal Study of Adolescent to Adult Health (Add Health), we construct both ordinal and Poisson regression models predicting poor self-rated health and the prevalence of chronic health conditions for both foreign-born and native-born groups, as well as different generations. The findings suggest important differences by nativity, immigrant generation, and type of justice system contact. Despite lower criminality than the native-born, the health of immigrants is deleriously impacted by some types of justice system contact, especially incarceration, while probation is more strongly linked to poor health among the native-born. Our findings carry implications for the provision of care for individuals with histories of criminal justice involvement, as well as academic research examining the consequences of justice contact and the immigration–crime nexus.

**Keywords:** immigration; criminal justice system; health; nativity; generation



## 1. Introduction

Both public and academic interest in how immigration shapes patterns of crime in the United States extends back nearly a century [1], with more rapid growth of such research over the past several decades. Driven by the most recent wave of foreign-born arrivals from Latin America and Asia [2], the proliferation of empirical scholarship has largely contradicted both political and public rhetoric of immigrant criminality. That is, immigrants appear to be less crime-prone than the native-born population [3,4], while communities with larger foreign-born populations tend to have equivalent or lower rates of crime than those experiencing less immigration [5], regardless of where immigrants arrive from [6]. However, there does appear to be some variation by the type of immigrant-receiving community [7–9].

Despite low levels of criminal offending, research demonstrates that immigrants exposed to the justice system experience a number of deleterious consequences. For example, parental immigration detention disrupts the educational trajectories of children [10]. Likewise, arrest and incarceration have been linked to poorer academic performance among children, declines in parent emotional wellbeing, poorer family financial resourcing, and strained familial relationships [11]. Not surprisingly, arrests have also been shown to reduce income and/or wages among the foreign-born [12]. Structurally, justice system involvement appears to exacerbate the vulnerability of immigrant populations, regardless of their lower levels of overall criminal involvement.

The current study extends this line of inquiry by examining two interrelated questions. First, we ask: is justice system contact (arrest, probation, and incarceration) linked to poorer health? Second, we ask: are there differences in whether justice system contact (arrest, probation, and incarceration) is associated with health outcomes (self-rated health and chronic conditions) among immigrant versus native-born persons? In doing so, we extend prior research on the role of criminal legal system involvement and health more generally, a literature that has grown tremendously in the past several decades, by examining disparities by nativity and immigrant generation. Additionally, the current study aims to describe the consequences of justice system involvement in a manner that would better direct the provision of health resources, as well as assist immigration advocacy work seeking to reduce the harmful systemic effects of justice contact among an already vulnerable population.

### 1.1. Justice System Contact, Stressors, and Health Outcomes

A considerable volume of empirical research reveals negative consequences linked to contact with the U.S. criminal justice system [13–15], including health and wellbeing. On the one hand, there appear to be some short-term health benefits associated with incarceration, at least somewhat attributable to access to care in carceral settings [16]. On the other hand, most empirical research finds that justice system contact (e.g., arrest, probation, and incarceration) is associated with poorer health. For example, justice system contact has been linked to poorer self-rated health, hypertension, depression, substance abuse disorders, chronic obstructive pulmonary disease, and physical and cognitive disability [17–21].

The dominant explanation for such findings centers on stress–process frameworks that see health disparities as the result of unequal exposure to structurally rooted stressors [22,23]. In essence, chronic exposure to stressors—including arrest or justice system punishment—results in vital regulatory systems (e.g., hypothalamic–pituitary–adrenal axis and the sympathetic nervous system) responding by releasing hormones to upregulate functioning across systems. Repeated overactivation of stress response systems leads these mechanisms to become inefficient, increasing the risk of disease susceptibility and early death [24]. Thus, justice system contacts act as stressors that undermine physiological regulation and deteriorate overall health.

### 1.2. Gaps in the Literature

Despite the proliferation of such health studies, two key gaps remain. First, and most germane to the current study, few studies have examined disparities by nativity or generational status (with at least one exception [25]). As we review, there are several reasons to suspect that foreign-born populations may differ in their risk of poor health resulting from justice system contact compared to other populations. Unfortunately, few empirical studies address these potential disparities despite prominent calls for such research [26], especially considering the low levels of criminality among the foreign-born.

Second, more research is needed that compares different types of justice system contact. Indeed, much research has been conducted on the role of incarceration on health generally, with comparably less research on earlier stages (e.g., arrest), alternative forms of punishment (e.g., probation), or the combination of punishments (e.g., incarceration and probation). As such, little is known about whether some types of justice system contact are more consequential for health than others, particularly for specific sub-populations such as the foreign-born broadly or specific immigrant generations. The goal of the current study is to address these gaps by examining specific health disparities in foreign-born versus native-born health as a possible function of different forms of criminal justice system contact, including differences across generation.

### 1.3. Health and the Foreign-Born: Prior Research and Theory

To date, research examining the role of justice system contact in shaping the health of immigrants has made important—but limited—strides. Much of it remains focused on immigrant deportation and detention or the threat of them. For example, there is some

evidence that fear of immigration enforcement is associated with increased stress and poor emotional wellbeing among foreign-born respondents [27]. Other studies observe significant increases in anxiety, depression, and post-traumatic stress disorder during and after detention among foreign-born people [28]. Additionally, immigration enforcement is linked to poorer self-rated health among both foreign- and native-born populations [29]. Some research even highlights the role of immigration detention in exacerbating infectious disease spread, particularly during the ongoing COVID-19 pandemic [30].

However, only a handful of studies directly examine broader justice system contacts and health outcomes while comparing foreign- versus native-born populations. The evidence of health disparities remains mixed. For instance, some research finds few differences across these groups in the effect of incarceration on self-rated health [25], whereas others similarly observe no difference in the relationship between personal and vicarious police contact and depression among both native- and foreign-born persons [31]. Likewise, there exists some evidence of fewer physical limitations with aging after justice system contact among the foreign-born compared to the native-born [32]. However, other research finds a greater prevalence of disease among foreign-born inmates compared to the native-born [33]. Taken as a whole, there remains a shortage of literature exploring the way justice system processes impact health among the foreign- versus native-born, though with mixed evidence that the foreign- and native-born differ in their health upon contact with the justice system.

Several prominent theoretical perspectives have been advanced to explain native- versus foreign-born health disparities. Indeed, a growing body of scholarship finds foreign-born populations exhibit better health outcomes [34,35], fewer chronic conditions [36,37], and more positive self-assessed health [38,39] than their U.S.-born counterparts. According to the healthy migrant hypothesis, observed health advantages among migrants may be due to their health profiles prior to migration. In other words, health makes migration more likely, such that migrants living in the U.S. are in better health than those who remain in their countries of origin or because better health is associated with other characteristics that predict migration, such as education or ability [40]. In a similar manner, the salmon bias hypothesis [41] argues that the foreign-born may return to their country of origin as a direct result of declining health [42] or that out-migration may be associated with other factors tied to poorer health [43].

Still, other theories posit that foreign-born populations engage in practices that are protective against poor health, including cultural traditions, accessing material and social support, and activating family ties that decrease the risk of poor mental and physical health [44,45]. Building from such perspectives, justice system contacts might have little or no impact on the health of the foreign-born given their existing health-buffering practices; that is, the factors that produce fewer health problems may similarly safeguard immigrants from the deleterious consequences of justice system contact. Indeed, immigrants—especially the first generation—may benefit from religious and cultural organization in residential ethnic enclaves that provide an "umbrella of social control" [46], a "shot of morality" [47], or resources (e.g., job placement, family assistance, medical care) that reduce the deleterious consequences of justice system contact on health. Likewise, immigrants may activate extended family networks that have been shown to offset individual disadvantages [48] in ways that also buffer against the health-related consequences of justice contact. However, whether there are, in fact, differences in the association between justice system and health among the foreign- and native-born (or across generations) remains an open empirical question, particularly given the gaps in the knowledge reviewed above.

## 2. Materials and Methods

### 2.1. Data

The current study seeks to address these gaps in knowledge relative to our two key research questions. To do so, data were drawn from Waves I–IV of the National Longi-

tudinal Study of Adolescent to Adult Health (Add Health), a nationally representative sample of over 20,000 adolescents in grades 7 to 12 who were attending 132 schools in 1994/1995 (Wave I). The school-based panel study is a random sample of all schools in the United States, stratified by region of the country, urbanicity, race/ethnicity, grade level, curriculum, and school type and size. Wave I included an in-school survey administered to every student in each school (N = 90,118), followed by an in-home interview (N = 20,745) of randomly selected students from the in-school survey. Wave IV interviews were administered in 2016–2018 and included 15,701 of the original Wave I respondents from the in-home survey. For inclusion in the analytic sample, respondents must have had valid responses for all measures described below, as well as valid sampling weights, resulting in a final sample size of 11,107. Most of the attrition in sample size from the 15,701 Wave IV respondents to our final sample size is due to missing sample weights (approximately 93 percent) or missing education, employment, or parental incarceration (which together account for the remaining 7 percent).

### 2.2. Dependent Variables

For this study, we included two health outcomes that capture both self-perception of the health and chronic conditions. Self-rated health remains one of the most consistently used indicators of physical health and is an established predictor of mortality [49]. Our measure of *poor self-rated health* was based on the following question: "In general, how is your health?" Responses ranged from 1 (excellent) to 5 (poor).

*Chronic conditions*, a count-based measure, capture the number of chronic conditions that have been diagnosed by a healthcare professional. The chronic conditions include cancer/lymphoma/leukemia, high blood cholesterol/triglycerides/lipids, high blood pressure/hypertension, high blood sugar/diabetes, heart attack/clogged coronary arteries, asthma/chronic bronchitis/emphysema, HIV/AIDS, hepatitis B or C, chronic kidney disease or failure, stroke/mini-stroke/clogged neck arteries, depression, post-traumatic stress disorder, and anxiety and panic disorder. This count-based measure of chronic conditions is widely used in social science scholarship [50,51] and has advantages over other measures that capture single health conditions (however, see the Robustness Checks section below) [52]. Both health outcomes are measured at Wave IV.

As an important caveat, we do not include lagged measures of health in our primary analysis. Unfortunately, our measure of chronic conditions is not assessed in prior waves, making it impossible to estimate models that include prior risk of chronic illness. At the same time, a supplemental model that includes prior poor self-rated health reveals substantively similar results (see Robustness Checks section below). Therefore, for the sake of consistency across our health outcomes, we present models that do not include prior health and suggest avenues for doing so in future research.

### 2.3. Key Independent Variables and Controls

Our history of criminal justice contact measures were derived from response items captured at Waves III and IV of the Add Health survey. At each of these waves, respondents were asked whether they had ever been arrested and, if they answered in the affirmative, then were asked about other types of criminal justice contact, including history of probation and history of incarceration. Using these items, we then created a measure that captures mutually exclusive types of criminal justice contact: (a) *no history of criminal justice contact*, (b) *arrest only*, (c) *probation only* (post arrest), (d) *incarceration only* (post arrest), and (f) *both incarceration and probation* (post arrest). Like other recent operationalizations of criminal justice contact [53], our measure represents Add Health respondents' exposure to the most severe level of criminal legal contact by Wave IV.

*Nativity* captures whether respondents were born outside of the U.S. or not (foreign-born versus native-born). *Immigrant generation* was measured using the country of birth of the respondents as well as the respondents' parents. Immigrant generation includes three categories: first-generation (foreign-born youth), second-generation (children of

immigrants), and third-generation plus. Adolescents categorized as first-generation were born in a foreign country, whereas those considered second-generation were born in the United States to at least one foreign-born parent. Adolescents who were born in the United States and whose parents (both) were also born in the United States were considered third-generation or more.

We also include a full set of controls. Gender was measured using a dummy variable for *female* (with male as the reference). *Age*, measured at Wave IV, was calculated as the difference between interview month and year and month of birth. Race was captured using dummy variables for non-Hispanic *White* (the reference category), non-Hispanic *Black*, and *Hispanic*. Education, also captured at Wave IV, was measured using three dummy variables for *less than high school*, *high school graduate*, and *more than a high school diploma* (reference). Income included four categories: *<USD 5000–USD 29,999, USD 30,000–USD 49,999, USD 50,000–USD 74,999,* and $\geq$ *USD 75,000* (reference) and was assessed at Wave IV. Family structure, derived from items included in the household roster at Wave IV, comprised four categories: *married with children*, *married without children*, *not married with children*, and *not married without children* (reference). Using codes from the Standard Occupational Classification System (SOCS), we included three work status categories: *not working*, *working in a non-professional job*, or *working in a professional job* (reference). Not working was defined as not having a job or working less than ten hours per week, while all occupations with SOCS prefixes 11–29 were defined as professional, and those with prefixes 31–55 were defined as non-professional. *History of parental incarceration* was assessed as to whether at least one parent had a history of incarceration. Finally, we created a binary measure indicating whether respondents had *health insurance* at the time of the Wave IV interview or not.

### 2.4. Analytic Strategy

We began by providing weighted descriptive statistics for the total sample and across nativity and immigrant generation. Following our descriptive analyses, we estimated a series of regression models to assess relationships between types of criminal justice contact and self-rated health and chronic conditions. For examining self-rated health, ordinal logistic regression was used. As chronic conditions are counts of those conditions and because diagnostic tests indicated that our chronic conditions measure exhibited no overdispersion, we estimated these models using Poisson regression (additional tests revealed no improvement in the model using negative binomial models, which provided nearly identical coefficients and standard errors).

To analyze the complex relationship between criminal justice contact, nativity/generation status, and health, we created models stratified by nativity and immigrant generation. Scholars have increasingly noted that using a within-group approach allows for isolation of stratification for historically marginalized groups, including immigrant populations, which will allow for more targeted policy interventions designed to address health inequalities. Although our analysis diverges from traditional approaches that examine relationships across immigrant groups (i.e., interaction models), preliminary analysis revealed no substantive differences in estimates using the within-immigrant group approach when compared to a multiplicative approach. Moreover, the advantage of the within-group approach is the estimation of all covariates simultaneously rather than through many interaction terms. All analyses account for the clustered nature of the Add Health sampling design as well as the unequal probability of selection due to oversampling based on racial/ethnicity, disability, and siblings, using poststratification weights.

## 3. Results

### 3.1. Descriptive Statistics

Table 1 presents weighted descriptive statistics for the total sample and by nativity and generation. The sample is overwhelmingly comprised of U.S.-born adults with two U.S.-born parents (third + generation, 85 percent), followed by the children of at least one

immigrant (second generation, 11 percent), and respondents that were born outside the U.S. (first generation, 4 percent). Moreover, the results indicate that almost one-fourth (23 percent) of respondents reported contact with the U.S. criminal justice system. Most respondents were female (56 percent), had at least a high school diploma (83 percent), were employed (66 percent), and were insured at the time of the Wave IV interview.

**Table 1.** Descriptive statistics for the total sample and by nativity and immigrant generation in Wave IV of the National Longitudinal Study of Adolescent Health.

| | Total | Foreign-Born | Native-Born | 1st Gen. | 2nd Gen. | 3rd Gen. |
|---|---|---|---|---|---|---|
| *Health Outcomes* | | | | | | |
| Poor self-rated health | 2.33 (0.01) | 2.17 (0.05) | 2.34 (0.01) | 2.17 (0.05) | 2.34 (0.03) | 2.34 (0.01) |
| Chronic conditions | 0.96 (0.01) | 0.51 (0.04) | 0.98 (0.01) | 0.51 (0.04) | 0.92 (0.04) | 0.98 (0.010) |
| *Immigration Measures* | | | | | | |
| Foreign-born | 0.04 (0.01) | | | | | |
| 1st generation | 0.04 (0.01) | - | - | - | - | - |
| 2nd generation | 0.11 (0.01) | - | - | - | - | - |
| 3rd + generation | 0.85 (0.01) | - | - | - | - | - |
| *Justice System Contact* | | | | | | |
| No contact | 0.77 (0.00) | 0.84 (0.02) | 0.77 (0.01) | 0.84 (0.02) | 0.80 (0.01) | 0.77 (0.01) |
| Arrest only | 0.06 (0.00) | 0.04 (0.01) | 0.06 (0.00) | 0.04 (0.01) | 0.05 (0.01) | 0.06 (0.00) |
| Probation | 0.05 (0.00) | 0.03 (0.01) | 0.05 (0.00) | 0.03 (0.01) | 0.05 (0.01) | 0.05 (0.00) |
| Incarceration | 0.04 (0.00) | 0.06 (0.01) | 0.04 (0.00) | 0.06 (0.01) | 0.03 (0.01) | 0.04 (0.00) |
| Prob. + incarc. | 0.08 (0.00) | 0.03 (0.01) | 0.08 (0.00) | 0.03 (0.01) | 0.07 (0.01) | 0.08 (0.00) |
| *Control Variables* | | | | | | |
| White | 0.73 (0.00) | 0.11 (0.02) | 0.75 (0.00) | 0.10 (0.02) | 0.43 (0.01) | 0.79 (0.00) |
| Black | 0.15 (0.00) | 0.06 (0.01) | 0.15 (0.00) | 0.06 (0.01) | 0.09 (0.01) | 0.16 (0.00) |
| Hispanic | 0.12 (0.00) | 0.84 (0.03) | 0.10 (0.00) | 0.83 (0.03) | 0.48 (0.02) | 0.06 (0.00) |
| Female | 0.56 (0.01) | 0.50 (0.03) | 0.56 (0.01) | 0.50 (0.03) | 0.55 (0.02) | 0.56 (0.01) |
| Age | 28.36 (0.02) | 28.99 (0.10) | 28.33 (0.02) | 28.99 (0.10) | 28.28 (0.07) | 28.34 (0.02) |
| Less than high school | 0.17 (0.00) | 0.16 (0.02) | 0.17 (0.00) | 0.16 (0.02) | 0.13 (0.01) | 0.18 (0.00) |
| High school | 0.41 (0.01) | 0.38 (0.03) | 0.41 (0.01) | 0.38 (0.03) | 0.40 (0.02) | 0.41 (0.01) |
| More than high school | 0.42 (0.01) | 0.46 (0.03) | 0.42 (0.01) | 0.46 (0.03) | 0.47 (0.02) | 0.41 (0.01) |
| Income < USD 5000-USD 29,999 | 0.21 (0.00) | 0.18 (0.02) | 0.21 (0.00) | 0.18 (0.02) | 0.18 (0.010) | 0.21 (0.01) |
| Income USD 30,000-USD 49,999 | 0.24 (0.00) | 0.20 (0.02) | 0.24 (0.00) | 0.20 (0.02) | 0.19 (0.01) | 0.24 (0.01) |
| Income USD 50,000-USD 74,999 | 0.26 (0.01) | 0.28 (0.03) | 0.26 (0.01) | 0.28 (0.03) | 0.28 (0.02) | 0.25 (0.01) |
| Income ≥ USD 75,000 | 0.24 (0.00) | 0.20 (0.02) | 0.24 (0.00) | 0.20 (0.02) | 0.19 (0.01) | 0.24 (0.01) |
| Married w/children | 0.29 (0.01) | 0.30 (0.02) | 0.29 (0.01) | 0.30 (0.02) | 0.28 (0.02) | 0.29 (0.01) |
| Married w/o children | 0.13 (0.00) | 0.18 (0.02) | 0.13 (0.00) | 0.18 (0.02) | 0.12 (0.01) | 0.13 (0.00) |
| Unmarried w/children | 0.16 (0.00) | 0.12 (0.02) | 0.16 (0.00) | 0.12 (0.02) | 0.16 (0.01) | 0.16 (0.00) |
| Unmarried w/o children | 0.42 (0.01) | 0.40 (0.03) | 0.42 (0.01) | 0.40 (0.03) | 0.45 (0.02) | 0.42 (0.01) |
| Not working | 0.34 (0.01) | 0.38 (0.03) | 0.33 (0.01) | 0.38 (0.03) | 0.30 (0.02) | 0.34 (0.01) |
| Non-professional job | 0.34 (0.01) | 0.30 (0.02) | 0.34 (0.01) | 0.30 (0.02) | 0.34 (0.02) | 0.34 (0.01) |
| Professional job | 0.32 (0.01) | 0.32 (0.03) | 0.32 (0.01) | 0.32 (0.03) | 0.36 (0.02) | 0.32 (0.01) |
| Parental incarceration | 0.13 (0.00) | 0.10 (0.02) | 0.14 (0.00) | 0.10 (0.02) | 0.14 (0.01) | 0.14 (0.00) |
| Currently insured | 0.82 (0.00) | 0.80 (0.02) | 0.82 (0.00) | 0.80 (0.02) | 0.86 (0.01) | 0.82 (0.00) |
| Language use | 0.03 (0.00) | 0.43 (0.03) | 0.02 (0.00) | 0.43 (0.03) | 0.14 (0.01) | 0.01 (0.00) |
| | *n* = 11,107 | *n* = 614 | *n* = 10,493 | *n* = 614 | *n* = 1454 | *n* = 9039 |

Central to the current study, there are key differences in health by nativity and immigrant generation. For example, Table 1 shows that the foreign-born fare better in poor self-rated health (2.17) than the native-born (2.34), as well as the number of chronic conditions (0.51 vs. 0.98). Though not shown, means comparison tests reveal that these differences are statistically significant ($p < 0.001$). Similarly, the first generation has a lower mean for both health outcomes ($p < 0.001$) than the second and third generations, the latter of which are statistically alike. Consistent with the prior literature [4], we also observe a rise in criminal legal contact with each immigrant generation. Specifically, 16 percent of respondents born outside the U.S. reported some form of criminal legal contact, fol-

lowed by 20 percent for the second-generation and 23 percent for third-generation (or higher) respondents.

*3.2. Regression Models*

Table 2 displays the results of the ordinal regression (poor self-rated health) and Poisson regression (chronic conditions) models examining the relationship between justice contact and health by nativity. For ease of interpretation, we display the criminal justice contact coefficients and standard errors for the criminal justice contact measures of arrest, probation, incarceration, and both probation and incarceration, though all models include a full set of controls as described above. The key findings are, first, that there are disparities in the association between justice system contact and both poor self-rated health and chronic conditions for foreign-born versus native-born respondents. On the one hand, justice contact has no relationship among immigrants for self-rated health, whereas a history of both probation and incarceration is associated with poorer self-rated health among native-born respondents. On the other hand, a history of incarceration only is associated with an increase in the number of chronic conditions among the foreign-born, while probation only is associated with more chronic conditions among the native-born.

Broadly, a second key finding from Table 2 is that different types of justice system contact appear linked to health in different ways. Notably, in comparison to no criminal justice contact, arrest by itself is unassociated with health outcomes across nativity groups. Instead, punishment stages of justice system contact are linked to poorer health outcomes, depending on nativity.

**Table 2.** Ordinal and Poisson regression models examining justice system contact on health by nativity among U.S. adults in Wave IV of the National Longitudinal Study of Adolescent Health.

| | Poor Self-Rated Health | | | | Chronic Conditions | | | |
|---|---|---|---|---|---|---|---|---|
| **Justice System Contact** | *Foreign-Born* | | *Native-Born* | | *Foreign-Born* | | *Native-Born* | |
| | **b (se)** | **OR** | **b (se)** | **OR** | **b (se)** | **IRR** | **b (se)** | **IRR** |
| Arrest only | 0.17 (0.64) | 1.19 | 0.03 (0.17) | 1.03 | −0.08 (0.68) | 0.92 | 0.09 (0.11) | 1.09 |
| Probation only | 0.61 (0.80) | 1.84 | 0.20 (0.21) | 1.23 | 0.81 (0.53) # | 2.24 | 0.30 (0.14) * | 1.35 |
| Incarceration only | 0.83 (0.78) | 2.29 | 0.18 (0.17) | 1.19 | 0.74 (0.34) * # | 2.10 * | 0.13 (0.14) | 1.14 |
| Incarceration + probation | −0.65 (0.92) # | 0.52 | 0.30 (0.14) * | 1.35 | 0.14 (0.68) | 1.15 | 0.04 (0.14) | 1.04 |
| | *n* = 614 | | *n* = 10,493 | | *n* = 614 | | *n* = 10,493 | |

No justice system contact serves as the reference group. All models control for age, race/ethnicity, education, gender, household income, language use, family structure, occupation, and parental incarceration. * $p < 0.05$, with # indicate statistically significant ($p < 0.05$) Wald tests for equality of coefficients between foreign- and native-born models [54].

Table 3 displays the same series of models estimated separately by immigrant generation (first generation, second generation, and third + generation) for both self-rated health and chronic conditions. As with our previous models, each is estimated separately with a full set of controls as described above. Our key finding here is that there are again differences in the association between justice system contact and both poor self-rated health and chronic conditions by immigrant generation. Among the first generation (the foreign-born), justice system contact is unassociated with poor self-rated health, whereas a history of incarceration by itself is associated with more chronic conditions, net of other controls. Among the second generation (native-born children of the foreign-born), justice system contact is also unassociated with poor self-rated health, but a history of probation is linked to a greater number of chronic conditions. Finally, among the native-born that are at least third generation, the combination of incarceration and probation histories is associated with poorer self-rated health but unassociated with chronic conditions. Thus, like our examination by nativity, different types of justice system contact are linked to health in different ways. Arrest by itself is unassociated with health outcomes across generational groups, whereas punishment is linked to poorer health outcomes, depending on the generation.

**Table 3.** Ordinal and Poisson regression models examining justice system contact on health by immigrant generation among U.S. adults in Wave IV of the National Longitudinal Study of Adolescent Health.

| | (A) Poor Self-Rated Health | | | | | |
|---|---|---|---|---|---|---|
| Justice System Contact | *1st Gen.* | | *2nd Gen.* | | *3rd + Gen.* | |
| | b (se) | OR | b (se) | OR | b (se) | OR |
| Arrest only | 0.17 (0.64) | 1.19 | −0.64 (0.49) | 0.53 | 0.08 (0.18) | 1.08 |
| Probation only | 0.61 (0.80) | 1.84 | 0.22 (0.29) | 1.24 | 0.25 (0.25) | 1.28 |
| Incarceration only | 0.83 (0.78) | 2.29 | 0.10 (0.89) | 1.10 | 0.19 (0.18) | 1.21 |
| Incarceration + probation | −0.65 (0.92) # | 0.52 | −0.21 (0.31) | 0.81 | 0.36 (0.17) * | 1.43 |
| | *n* = 614 | | *n* = 1454 | | *n* = 9039 | |
| | (B) Chronic Conditions | | | | | |
| Justice System Contact | *1st Gen.* | | *2nd Gen.* | | *3rd + Gen.* | |
| | b (se) | IRR | b (se) | IRR | b (se) | IRR |
| Arrest only | −0.08 (0.68) | 0.92 | −0.01 (0.41) | 0.95 | 0.11 (0.11) | 1.11 |
| Probation only | 0.81 (0.53) # | 2.24 | 0.89 (0.23) *** | 2.14 | 0.19 (0.12) | 1.21 |
| Incarceration only | 0.74 (0.34) * # | 2.10 * | −0.23 (0.32) | 0.75 | 0.15 (0.15) | 1.17 |
| Incarceration + probation | 0.14 (0.68) | 1.15 | | 0.68 | 0.07 (0.14) | 1.08 |
| | *n* = 614 | | *n* = 1454 | | *n* = 9039 | |

No justice system contact serves as the reference group. All models control for age, race/ethnicity, education, gender, household income, language use, family structure, occupation, and parental incarceration. * $p < 0.05$, and *** $p < 0.001$ with # indicate statistically significant ($p < 0.05$) Wald tests for equality of coefficients between generation groups (any statistically significant difference indicated).

*3.3. Robustness Checks*

In order to assess the robustness of our findings, we conducted several sensitivity checks including (a) constructing a model predicting poor self-rated health with the inclusion of a lagged measure of poor self-rated health; (b) estimating models separately predicting groups of chronic conditions (e.g., infectious disease, mental health, and other chronic conditions); (c) replacing our measure of occupational status with a binary employment measure; and (d) estimating our multivariable models with the inclusion of cases that are missing appropriate weights. Generally, the pattern of findings remains unchanged. For example, the inclusion of lagged poor self-rated health did not attenuate the association between the combination of incarceration and probation with poor self-rated health for the native-born. Likewise, the substitution of employment for occupational status and the estimation of models that include all cases with missing weights resulted in no change to the overall pattern of relationships between justice system contact and our health outcomes. Finally, the supplemental models disaggregating chronic conditions revealed that the patterns observed for the overall aggregate models displayed in Tables 2 and 3 were driven by a combination of mental health and non-disease chronic conditions more than infectious illnesses (i.e., HIV/AIDS and hepatitis). This may reflect that few respondents report such chronic conditions relative to the more ubiquitous mental and physical health conditions also included in the chronic conditions index. Overall, our findings appear robust to alternative specifications.

**4. Discussion**

The goal of the current study was to address whether justice system contact (arrest, probation, and incarceration) was linked to poorer health and, in turn, whether there were differences in those relationships between immigrant versus native-born persons. Broadly, our study was informed by growing empirical attention on the health impact of justice system contact, as well as prominent theories of immigrant health advantage. In particular, the current study aimed to address two key gaps in prior research with few studies comparing across types of justice contact or with attention centered on immigrant health as it relates to different histories of justice system exposure.

We identified three broad findings. First, our results suggest that criminal justice contacts are one of many factors affecting health outcomes and, generally, there are more similarities than differences across nativity and generation. Other familial, community, and personal factors are likely to be central to understanding health disparities, and only some types of late-stage justice contact are associated with disparities in health. Thus, what happens after adjudication among immigrants (e.g., the activation of social networks or access to resources that buffer against negative health consequences) may be especially important, but the health consequences of justice contact are generally small.

Second, per our first research question, where we found that some justice system contact was associated with both poor self-rated health and chronic conditions, there was little indication that histories of arrest alone were associated with health outcomes. Instead, punishment (probation and/or incarceration) was linked to poorer health outcomes for both self-rated health and the number of chronic conditions. This finding marks an important contribution to the existing literature on both health disparities and the consequences of criminal justice contact by demonstrating the importance of including incarceration and alternative punishments (e.g., probation), both which were found to be most strongly linked to deleterious health outcomes. Thus, the health consequences of punishment are not only experienced with incapacitation; they result from more common forms of carceral control that, despite their ubiquity, are often viewed as "less serious" than incarceration.

As a third key finding, we observed some disparities by nativity and generation that, though few, reflect important disparities in the health consequences of justice system contact. For the foreign-born, justice contact was unassociated with self-rated health, but incarceration was linked to more chronic conditions. For the native-born, the combination of probation and incarceration histories were linked to poorer self-rated health, whereas probation was associated with more chronic conditions. However, these broad patterns among the native-born obscured important generational disparities: the health of the children of immigrants (second generation) was most impacted by probation, while the health of their children (third generation) was most linked to the combination of probation and incarceration histories. Thus, relative to our second research question, justice system contact was not associated with health outcomes uniformly by nativity or immigrant generation.

Such findings make important contributions to the extant immigration crime literature. Much prior criminological work has shown that foreign-born persons are no more prone to crime than the native-born [3,4], and the relative size of the immigrant population in a community generally has no association with crime [5], though with some regional/community-type differences [7–9]. The current study demonstrates that justice system contact remains consequential for immigrants *despite lower levels of criminality*. Clearly, the immigrant health advantage does not extend in uniformly protective ways to those foreign-born persons who have contact with the justice system.

Our findings also matter for the provision of healthcare both within the justice system, as well as after contact for those who re-enter the community. For one, not all forms of justice system contact appear consequential for health, and, as such, care should be taken to focus scarce resources on individuals for whom contact matters most. In turn, results from the current study suggest that native-born and immigrant populations face different health consequences with justice system contact and that models of care should similarly vary. Whereas native-born populations appear most impacted by the combination of incarceration and probation relative to more immediate (self-rated) health, foreign-born persons who have been incarcerated may be in greater need of care for chronic conditions (e.g., cancer/lymphoma/leukemia, high blood cholesterol, hypertension, etc.). Like most inequalities, solutions are not "one size fits all."

Despite what we see as important contributions to the growing immigration, crime, and justice literature, there are important ways to extend the current study to better inform research and practice. For example, justice contact was captured in the current study in ways that (a) combine warranted versus unwarranted contact with the justice system; (b) do not capture other stages of justice system processing, such as conviction, parole, or

immigrant-specific detention; and (c) do not capture the number of justice system exposures (or duration). There are good reasons to suspect important nuance in the ways that criminal justice contacts shape health for vulnerable populations, such as the foreign-born. Such limitations of the current study could certainly prove fruitful avenues of empirical discovery and better direct policymakers and practitioners in ways to reduce health disparities within and outside of the justice system.

Moreover, the role played by justice system contact for health may be more particular than the broad measures examined here, especially when comparing across immigrant groups. Indeed, our own supplemental models revealed some differences in how different types of justice system involvement were linked to different types of chronic conditions. More research will be needed that fully contextualizes such key relationships. Similarly, we were unable to fully account for prior health, and more work will be needed that examines the longitudinal trajectories of wellbeing amidst different types of justice system contact. Finally, we note that Add Health is only representative of students enrolled in school and that our findings may not be generalized to individuals who may have dropped out or to immigrants who migrated at the age when most schooling is generally completed.

## 5. Conclusions

Although immigration may not have a criminogenic association with crime and immigrants may be less crime-prone than suspected by the public, there remain important consequences for those immigrants that are in contact with the justice system, including for their health.

**Author Contributions:** Conceptualization, C.T.H. and M.N.; methodology, M.N.; formal analysis, M.N.; investigation, C.T.H.; resources, C.T.H., M.N. and M.R.; data curation, M.N.; writing—original draft preparation, C.T.H., M.N., Z.Z. and M.R.; writing—review and editing, C.T.H., M.N., Z.Z. and M.R.; supervision, C.T.H. and M.N.; project administration, C.T.H., M.N., Z.Z. and M.R. All authors have read and agreed to the published version of the manuscript.

**Funding:** This research received no external funding.

**Institutional Review Board Statement:** The study was conducted in accordance with the Declaration of Helsinki and approved by the Institutional Review Board (or Ethics Committee) of Willamette University (protocol code 00006782 on 7 October 2019).

**Informed Consent Statement:** Informed consent was obtained from all subjects involved in the study.

**Data Availability Statement:** The data presented in this study are openly available at the National Longitudinal Study of Adolescent Health curated by the University of North Carolina at https://addhealth.cpc.unc.edu/data/#public-use, accessed on 30 September 2022.

**Conflicts of Interest:** The authors declare no conflict of interest.

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
