# Peer review of "Justice System Contact and Health: Do Immigrants Fair Better or Worse than the Native-Born after Arrest, Probation, or Incarceration?"

_societies, doi:10.3390/soc13030077_

Round 1
Reviewer 1 Report
Please see attached comments.

Author Response
February 26, 2023
We want to thank the reviewers and editors for taking the time to carefully evaluate our manuscript. We were pleased that each reader saw value in the manuscript and, with careful revisions, would make a valuable contribution to the literature.
As described below, we were able to address every one of these suggested revisions. In turn, we believe the manuscript is significantly improved. Each reviewer comment is denoted in bold text, with our responses in normal font. All changes in the manuscript were noted using tracked changes, as requested by the editors.
The two reviewers listed the following as major revisions:
- Add justification for the current study within section 1.3 by elaborating on the theoretical expectations. Reviewer #1 asks that we further describe why we would expect differences by nativity and immigrant generation in the association between justice system contact and health. Similarly, Reviewer #2 asks that we flesh out the theories a little bit more. We see these two points as complementary and address them jointly below.
We agree with the reviewers that expectation of disparities in the justice contact-health relationship remain underdeveloped. To that end, we expanded considerably (and added citations to support) our discussion on page 3 to outline theoretical frameworks positing that foreign-born populations engage in practices that are protective against poor health, including cultural traditions, accessing material and social support, and activating family ties that decrease the risk of poor mental and physical health. Building from such perspectives, justice system contacts might have little or no impact on the health of the foreign-born given their existing health-buffering practices – that is, the factors that produce fewer health problems may similarly safeguard immigrants from the deleterious consequences of justice system contact. Indeed, immigrants – especially the first generation – may benefit from religious and cultural organization in ethnic enclaves that provide an “umbrella of social control”, a “shot of morality”, and resources (e.g., job placement, family assistance, medical care) that reduce the deleterious consequences of justice system contact on health. Likewise, immigrants may activate extended family networks that have been shown to offset individual disadvantages in ways that also buffer against the health-related consequences of justice contact. However, whether there are, in fact, differences in the association between justice system and health among the foreign- and native-born (or generation) remains an open empirical question, particularly given the gaps in knowledge reviewed above.
- Describe the data in more detail. Reviewer #1 asked that we “provide more information about [the] sample selection criteria…[including] what percentage is due to the sample weights versus listwise deletion, as well as provide more clarity on the use of Wave III data within the 2.1 Data section of the manuscript (i.e., what variables are taken from the wave?).
Thank you for the helpful suggestion. We have done this on pages 4 (of 13) where we now note that missingness was overwhelmingly due to omitted sampling weights with 93 percent of those cases dropped from the Wave IV analytic sample due to such omission (the remaining 7 percent is attributable to missingness on education, employment, and parental incarceration). Nevertheless, we also included on page 9 in our new Robustness Checks section that the inclusion of all cases ended up producing nearly identical patterns in our findings relative to the justice system contacts and their association with health.
Moreover, we have now clarified on page 4 that the criminal justice contact measures were taken from both Wave III and Wave IV. We apologize for this omission in the prior manuscript where we noted the use of Wave III but did not mention that it is used to construct just these justice-related variables.
- Clarify the modeling strategy. Reviewer #1 also notes that we employed diagnostic tests indicating overdispersion, but that those tests revealed no improvement in model using negative binomial forms. However, the reviewer notes that overdispersion is generally the criteria used to select negative binomial models over Poisson models because equi-dispersion is a central assumption of Poisson, so this is a confusing justification.
Thank you for catching this error. We agree that there was confusion about the functional form of our data as described on page 5. We have revised the manuscript to note that our chronic conditions measure is count-based but not over-dispersed. As such, we use Poisson regression models and display the unstandardized coefficients and standard errors alongside incident rate-ratios (for ease of interpretation) in both Table 2 and Table 3. As further evidence, we also note that negative binomial models produce nearly identical coefficients and standard errors, as would be expected with no model fit improvement.
- Add unstandardized logit coefficients (or add confidence intervals) and coefficient comparison tests to the tables. Reviewer #1 asked that we add logit coefficients (or confidence intervals) to better contextualize the standard errors. Likewise, they note that the comparisons we articulate as a primary goal of the manuscript should be accompanied by appropriate statistical tests (Wald) for the equality of those relationships across samples (and very thankfully provided a citation to do so). Similarly, Reviewer #2 echoes the call for coefficient comparisons given that we are not presenting multiplicative models with many interaction terms.
Consistent with the suggestion of both reviewers, we have revised Tables 2 and 3 to include the unstandardized coefficients and their corresponding standard errors, as well as the odds ratios (for the ordinal regression models) and incident rate-ratios (for the Poisson regression models). In doing so, we have moved the significance indicators to these coefficients and cleaned up the organization of the tables to fit the journal format and spacing. Given these additions and the limited space available for each table, we felt that it was unnecessary to present the confidence intervals since the coefficients and standard errors can be used to more easily contextualize the relationships observed in each table. We would be happy to add them should the editors request us to do so.
- Consider including a measure of prior health. Reviewer #2 suggests a sensitivity analysis that includes a control for baseline health.
This is a great suggestion, though one we cannot fully address given data constraints in the current manuscript. Most central to the critique, a comparable measure of chronic conditions is not included in prior waves of these data, making it impossible to include such a lagged measure. In addition, we estimated supplemental models exploring the inclusion of a lagged poor self-rated health measure and found that the association of criminal justice contacts with subsequent poor self-rated health were largely unchanged. This may reflect the fact that very few types of contact were related to poor self-rated health in the unlagged models to begin with. Nevertheless, we added a comment on page 4 (of 13) to note that we do not include lagged health measures for these reasons, as well as the supplemental model on pages 9-10 where we describe other alternative models to check the robustness of our findings.
- Address whether the models require controls for race/ethnicity. Reviewer #2 was “very surprised to see that the authors do not control for race and ethnicity in their analysis.”
We apologize for this oversight and assure the reviewer that we did in fact include these in our models, but failed to include them in the table or a description of our variables. Indeed, the discussion of key variables was based upon Table 1 and, thus, omitted these variables, as well. To be clear, we have now added this material on page 5 (of 13) where we note our models include controls for non-Hispanic Black and Hispanic with non-Hispanic White serving as the reference category. Again, we apologize for the confusion this caused.
- Consider a sensitivity check by disaggregating chronic conditions. Reviewer #2 asks us to consider a sensitivity analysis of individual chronic health items, or at least differentiate between groups of chronic illnesses (e.g., infectious, mental health, and other diseases).
This is an excellent point and one which we expanded to address both reviewers’ comments throughout their review letters. Specifically, we created a Robustness Checks section that describes how we estimated supplemental models disaggregating or grouping chronic conditions into infectious (HIV/AIDS, hepatitis), mental health, and then other chronic conditions (see pages 9-10). We now note that patterns observed for the overall aggregate models displayed in Tables 2 and 3 were driven by a combination of mental health and non-disease chronic conditions more than infectious illnesses (i.e., HIV/AIDS, hepatitis). That is, we found that the link between justice contact and health was less about infectious conditions than more common, physical and mental ailments. This may reflect that few respondents report infectious conditions relative to the more ubiquitous mental and physical health conditions also included in the chronic conditions index.
- Add formal comparisons of the health outcomes to the descriptive statistics table. Reviewer #2 asks that we provide a significance test in Table 1 that could serve as an early (bivariate) comparison of whether there are observed health disparities by nativity or generation.
We have now added on page 6 (of 13) that there are key difference in health by nativity and immigrant generation. For example, Table 1 shows that the foreign born fare better in poor self-rated health (2.17) than the native born (2.34), as well as the number of chronic conditions (.51 vs. .98). Though not shown, means comparison tests reveal that these differences are statistically significant (p<.001). Similarly, the first-generation have a lower mean for both health outcomes (p<.001) than the second and third generation, the latter of which are statistically similar to each other.
- Clarify the employment measure. Reviewer #1 asks why we use our specific employment measure rather than a binary of employed vs. unemployed.
This is an interesting point which we addressed by constructing another supplemental model (see the Robustness Checks section). Our initial use of occupational status reflected the assumption that health outcomes are affected by both employment, as well as the type of work, which can determine access to health-related resources (e.g., insurance). Thus, we chose to use a measure of occupational status rather than a simple binary of employment. Nevertheless, we estimated a supplemental model and the overall pattern of our results does not differ.
- Temper the discussion a bit to acknowledge the nuances of the results. As Reviewer #1 notes, only three effects are significant in each table, suggesting that there is more similarity in health outcomes than perhaps the authors anticipated. They ask that we spend a bit more time reflecting on such patterns.
This is another excellent point and we have added language on page 10 that notes that there are more similarities than differences across populations in our study. While there are some disparities (and we continue to describe them on pages 10-11), we first note that our findings suggest that criminal justice contacts are one of many factors affecting health outcomes and, generally, there are more similarities than differences across nativity and generation. Other familial, community, and personal factors are likely to be central to understanding health disparities and only some types of late-stage justice contact are associated with disparities in health. Thus, what happens after adjudication among immigrants(for example, the activation of social networks or access resources that buffer against negative health consequences) may be especially important, but the health consequences of justice contact are generally stable.
The two reviewers also listed several minor revisions that we have also fully executed. These include:
- Note that Add Health is only representative of students enrolled in school. In turn, that means the findings cannot be generalized to individuals who may have dropped out and to immigrants who migrated at the age when most of schooling is generally completed. This is now included on page 11.
- Revising the language on Wave IV of Add Health. As the reviewer points out, Wave V of Add Health is now available, and that Wave IV is no longer the most recent. We simply deleted reference to “the most recent” for Wave IV.
- Clarify the generation measure. We mistakenly included the phrase “foreign born to at least one parent” when describing the first generation. We have deleted it and instead refer to generations as follows: “Adolescents categorized as first generation were born in a foreign country, whereas those considered second generation were born in the United States to at least one foreign-born parent. Adolescents who were born in the United States and had both parents also born in the United States were considered third generation or more.”
- Proofreading. Both reviewers caught several typos and grammatical errors. We have carefully read through the document several times to ensure errors are minimized.
We would again like to thank the reviewers and editors for their helpful comments. We look forward to a continually constructive review process.
Regards,
The authors

Reviewer 2 Report
Thank you for the opportunity to review this paper. I enjoyed reading it. Overall, this study does seek to fill important gaps in immigration and justice as well as in health consequences of CJ-system contact research. The paper is generally well written and includes a good overview of relevant literature (although see my comment regarding the theoretical framework and editing suggestions at the end). However, as is, I believe the analysis fails to properly test the hypotheses stated by the authors and other parts of the paper need to be strengthened before I can recommend publishing this study. I think most of my concerns can be addressed, so I recommend an R&R.
I would like to first discuss the issues that are most concerning to me and then provide minor suggestions section-by-section.
Main Concerns
First, I think the main contribution of this paper is in examining whether CJ system contact affects health outcomes of foreign born and native born (and by generation) differently. Yet, the authors did not conduct any formal statistical tests to compare the coefficients in the models disaggregated by nativity and generation, such as the commonly used Paternoster et al. (1998) test. Instead, the authors rely on statistical significance versus the lack thereof of coefficients in the subsamples to identify differences in the impact of CJ on health, which amounts to informal and likely erroneous eyeballing approach. I know the authors also examined multiplicative models, where the significance of interaction effects can be used for the same purpose. However, they do not provide the results of these models, so it is not clear whether the impact of CJ contact on health outcomes differs by nativity or generation. Please provide a formal test of equality of coefficients in the subsamples.
Second, to properly test whether CJ system contact impacts health outcomes, the models really need to include some kind of measures of health prior to the contact with CJ system. I know this would substantially reduce the amount of unexplained variation in the outcome and might make the effects of other variables not significant, but at least a sensitivity analysis that includes some kind of control for baseline health is advisable.
Third, (and I genuinely hate to comment on control variables because I know that adding them forces authors to re-do all of the regression tables often without much substantive change in the results) I am very surprised to see that the authors do not control for race and ethnicity in their analysis. I think in this case this can lead to serious spuriousness issues. For example, in Table 2, maybe the reason Incarceration+Probation impacts health of native born and not of foreign born is because the native-born group may include a greater proportion of Black respondents who maybe at a great risk of sanctions and experience additional health issues associated with the stress stemming from racial discrimination. I think the analysis must include measures of race and ethnicity.
Those are my major concerns. The following are more minor issues, section-by-section…
Literature Review
I imagine there are space limitations imposed by the journal. However, I think the theoretical framework should be strengthened. While “selective migration” and “salmon bias” are important to mention for context, the study hypotheses are focused on how CJ system contact affects health after the migration has already occurred. The authors do note that “… theories posit that foreign-born populations engage in practices that are protective against poor health, including cultural traditions, accessing material and social support, and activating family ties …” which seems much more relevant, but this is limited to a two sentences that talk about this at a great level of generality. I recommend spending a little less time talking about the selection effects and flesh out the theories mentioned in the statement above a little bit more. I think that could also help strengthen the interpretation of the study results later in the paper.
Data
The authors state that “Most of the attrition in sample size is the result of missing sample weights”. While sample weights are important in generating descriptive statistics that are representative of the population, they tend to have less impact on the relational model results (i.e., regression). I suggest doing a sensitivity test by re-running the regression models ignoring the weights to see if this would substantially affect the results.
DV
While the authors provide good justification for using a composite measure of chronic conditions (and cite studies supporting this), to me these conditions are so substantively different that I think that a sensitivity analysis of individual measures, or at least differentiating between groups of chronic illnesses like infectious, mental health, and other diseases is worth a look.
Results
Descriptive statistics should include a statistical significance test, at least for the two outcome measures by foreign/native and by generation. I know interpreting the differences without adjusting for control variables is risky, but most studies start with bivariate hypotheses test before moving on to multivariate models. Providing a significance test in Table 1 could serve as this baseline bivariate test of the hypotheses. It would be interesting to see if the initial between group differences in health outcomes disappear in multivariate models.
While I agree that ORs and IRR are easier to interpret, I think the authors should provide raw coefficients, raw standard errors and OR and IRR in tables 2 and 3 in case this study ends up being included in someone’s meta-analysis.
Limitations
The authors should note that Add Health is only representative of students enrolled in school and that the findings cannot be generalized to individuals who may have dropped out and to immigrants who migrated at the age when most of schooling is generally completed.
Authors should also note that the measures of CJ contact do not measures duration of the contact.
Editorial Stuff
Page 5, line 209, 2010 “Because chronic conditions are by counts of those conditions” I think should be “measured by count…”
Page 5 214, 215 “Finally, it is important to note that all regression models included in this study 214 were estimated by nativity and immigrant generation.” – Statement seems unnecessary since this is explained in the next paragraph.
Page 8, line 341 “In this way,” sentence cuts off
References
Paternoster, Raymond, Robert Brame, Paul Mazerolle, and Alex Piquero. 1998. Using the Correct Statistical Test for the Quality of Regression Coefficients. Criminology 36(4):859-866
Author Response

(The authors gave the same response as above.)

Round 2
Reviewer 1 Report
I am very pleased with the authors' revisions and responses to reviewer comments. There are only two remaining points I think the authors should address.
First, I appreciate the authors' adding the results of Wald tests for the equality of coefficients. However, the way they are presented in the tables is a bit confusing. I recommend clarifying the comparison further in the table notes. As it reads now, it sounds like the #s reflect coefficients that are equal, rather than ones that are not. It could be revised to something like "# indicating that coefficient for foreign born is significantly different (p < .05) from corresponding coefficient for native born, based on Wald tests." Further, but relatedly, it would be better if these Wald tests in Table 3 were tests between each comparison, rather than overall. For example, there is a # next to the coefficient for Probation among 1st gen, indicating that the effect of Probation varies by generational status. But what is the nature of that variation? Are all three coefficients different? Or is 3rd gen different from 1st and 2nd, but 1st and 2nd are the same? I know this may seem minor, but it is an important distinction for interpretation.
Second, the only comment from my initial review that the authors were not responsive to was my suggestion that the authors not refer to third generation respondents as the children of second generation respondents (now page 10) because they are referring specifically to the patterns in their data among respondents and none of the respondents are children of other respondents. The sentence currently reads: "the health of the children of immigrants (second generation) were most impacted by probation, while the health of their children (third generation) were most linked to the combination of probation and incarceration histories." Again, the third generation respondents are not "their" children.
Reviewer 2 Report
I believe that the revisions the authors made addressed all of the substantive issues I identified in my first review. I think maybe one more round of proof-reading would be a good idea, but this can be done in the copy-edit stage. I do not have any additional suggestions. I think the manuscript is ready for publications. Thank you for inviting me to review this study.